# Bioenergetic trade-offs can reveal the path to superior microbial CO$_2$ fixation pathways

Ahmed Taha,[1] Mauricio Patón,[1] Jorge Rodríguez[1]

**ABSTRACT** A comprehensive optimization of known prokaryotic autotrophic carbon dioxide (CO$_2$) fixation pathways is presented that evaluates all their possible variants under different environmental conditions. This was achieved through a computational methodology recently developed that considers the trade-offs between energy efficiency (yield) and growth rate, allowing us to evaluate candidate metabolic modifications *in silico* for microbial conversions. The results revealed the superior configurations in terms of both yield (efficiency) and rate (driving force). The pathways from anaerobic organisms appear to fix carbon at lower net ATP cost than those found in aerobic organisms, and the reverse TCA cycle pathway shows the lowest overall energy cost and maximum adaptability across a broad range of CO$_2$ and electron donor (H$_2$) concentrations. The reverse tricarboxylic acid cycle and Wood-Ljungdahl pathways appear highly efficient under a broad range of conditions, while the 3-hydroxypropionate 4-hydroxybutyrate cycle and the 3-hydroxypropionate bicycle appear capable of generating large thermodynamic driving forces at only moderate ATP yield losses.

**IMPORTANCE** Biotechnology can lead to cost-effective processes for capturing carbon dioxide using the natural or genetically engineered metabolic capabilities of microorganisms. However, introducing desirable genetic modifications into microbial strains without compromising their fitness (growth yield and rate) during industrial-scale cultivation remains a challenge. The approach and results presented can guide optimal pathway configurations for enhanced prokaryotic carbon fixation through metabolic engineering. By aligning strain modifications with these theoretically revealed near-optimal pathway configurations, we can optimally engineer strains of good fitness under open culture industrial-scale conditions.

**KEYWORDS** prokaryotic carbon fixation, carbon metabolism, electron carrier analysis, rate-yield trade-off, microbial strain engineering, microbial thermodynamics, optimisation of microbial pathways

I n the last century, the concentration of atmospheric carbon dioxide (CO$_2$), a crucial component of the natural carbon cycle (1), has been increasing with detrimental climate and ecological effects (2). This has brought economic and technological interest in its removal from the atmosphere by intensified capture and fixation (3). The development of sustainable processes to not only capture and fix CO$_2$ but also to transform it into economically valuable end products is of major interest. Chemical processes already exist today for those goals, such as Fischer-Tropsch synthesis. However, they are energy-intensive processes requiring high temperatures and/or pressures, or previous transformation steps of part of the CO$_2$ into CO (4). Biotechnological approaches based on the use of the metabolic capabilities of microorganisms bring about sustainable alternatives to high-intensity processes to achieve the same goals at lower cost.

**Peer Reviewer** Christopher E. Lawson, University of Toronto, Toronto, Canada

Address correspondence to Jorge Rodríguez, jorge.rodriguez@ku.ac.ae.

The authors declare no conflict of interest.

The engineering of microbial communities has a long history of successful top-down approaches, especially in the field of wastewater treatment (5, 6). These approaches do however tend to neglect the complex metabolic networks driving microbial and chemical transformations as well as specific intricate interactions between community members (e.g., syntrophic interspecies electron transfer) (7). On the other hand, the recent advances in molecular techniques have enabled a much deeper understanding of the vast range of microbial metabolic capabilities present in wastewater treatment plants (8) and allowed for the development of bottom-up approaches to engineer microbial community metabolic networks. Bottom-up methods, such as flux balance analysis, provide quantitative models, in which individual reactions and metabolic fluxes within and between populations can be simulated based on the application of optimality principles. There are numerous computational tools to systematically evaluate metabolic networks of interest (7). This bottom-up engineering of microbial ecosystems seeking to optimize the ecosystem function and stability has been explored in wastewater treatment through specific microbial enrichments (9) or by direct metabolic engineering of microbial strains (10). Genetic modification of microbial strains has also been explored as an alternative for carbon fixation (11). Flux balance analysis is, however, applied typically to simple communities with model organisms; extending its application to more complex high-diversity microbial systems will necessarily require better models incorporating other applicable natural principles such as energetic considerations.

The trade-offs between metabolic energy efficiency (impacting on yield) and driving force (impacting on rate) are critically important to the fitness of any engineered strain in a mixed microbial culture. This is especially true for microbial growth in environments under energy-scarcity selection forces such as in anaerobic wastewater treatment and other fermentative bioprocesses of interest. Introducing desired metabolic modifications into a microbial strain or community that are effective and have lasting viability is challenging unless they have a positive, or at least not detrimental, impact on fitness. This can be achieved if the modifications are aligned with ecologically valuable traits such as enhanced growth rate. Finding desired engineered modifications that are also in line with optimality in terms of growth is therefore of the maximum interest. Maximum growth rate, especially in energy-limited systems, lies on the trade-off front between high energy efficiency (yield) and sufficiently large driving forces (rate). Understanding the impact of environmental or operational conditions on these trade-offs can help bridge the gap between top-down and bottom-up approaches in microbial community engineering.

The biological fixation of inorganic carbon (carbon dioxide in any of its water dissolved forms $CO_{2(aq)}$, $H_2CO_3$, $HCO_3^-$, and $CO_3^{2-}$) involves its transformation into organic compounds that can be incorporated in the metabolism of living cells. Carbon fixation into biomass building blocks requires a reduction from inorganic carbon (its most oxidized form) into organic carbon by incorporation of circa 4.2 electrons per carbon in typical biomass (12, 13). In non-photosynthetic carbon fixation, the reducing equivalents required for this stepwise process are sequentially supplied by donating electron carriers (e.g., NADH, $FADH_2$) at different energy levels (reduction potentials). Any energy required for the carbon fixation reduction steps in non-phototrophic organisms originates either from the cell's ATP/chemiosmotic proton motive force pool or from direct coupling with the electron donor reaction itself (14). The majority of reported autotrophic carbon fixation pathways lead to either acetyl-CoA (two-carbon) or pyruvate (three-carbon), both central metabolites and building blocks for multiple other subsequent compounds toward the synthesis biomass or other metabolic products (15). Biomass itself is a medium-value target product together with other platform chemicals building blocks toward higher-value products such as solvents and fuels, generally more reduced and longer in carbon length (16).

In nature, the majority of carbon fixation occurs photosynthetically by oceanic phytoplankton and other members of the Plantae kingdom via the Calvin cycle metabolic pathway (17). Photosynthetic carbon fixation is however limited by the

availability of light and the inherent inefficiencies of photosynthesis, which prevent its scale-up at low cost. Several alternative dark (light-independent) microbial carbon fixation pathways of potential relevance for process development exist. At least six main pathways are reported for autotrophic assimilation of $CO_2$ into biomass to date (14, 15, 18–37). The complete detailed descriptions of these pathways are presented in Table 1 and other figures in Materials and Methods as well as in Supplemental Material S1 in full.

Apart from the biochemical feasibility, in order for a given pathway to proceed, it must also be thermodynamically and kinetically feasible. An available approach to link kinetics to thermodynamics is the framework developed by Noor and colleagues (38) that seeks to maximize the minimum (smallest) driving force of all elementary reaction steps in a pathway, the driving force being the Gibbs energy dissipated in each reaction. The rate of a given biochemical reaction step can be coupled to its driving force by defining a so-called flux-force efficacy (FFE), an enzyme utilization factor calculated as per equation 1:

$$cFFE = \frac{e^{\frac{\Delta G_{diss}}{RT}} - 1}{e^{\frac{\Delta G_{diss}}{RT}} + 1} = \frac{J^+ - J^-}{J^+ + J^-} \tag{1}$$

where $J^+$ and $J^-$ are the forward and backward rates of the reaction step, respectively, $\Delta G_{diss}$ is the reaction's Gibbs energy dissipated (the negative of the Gibbs energy change of the reaction, a quantity that represents the reaction's driving force), $R$ is the ideal gas constant, and $T$ is the absolute temperature. According to this model, under very small driving forces, a reaction is close to equilibrium and much of the enzyme activity is occupied by the catalysis of the backward flux, thus effectively lowering the maximum net reaction rate. At larger driving forces, however, the reaction is away from equilibrium, increasing the FFE and improving the enzyme utilization efficacy; this translates into either a higher net forward rate, or a comparable rate requiring a smaller quantity of enzyme. Considering that the overall rate of a sequential pathway is limited by the slowest reaction step, the maximization of the minimum (smallest) driving force (MDF) targets the potentially rate-limiting step increasing the overall pathway rate. Several studies that focus on the thermodynamic analysis of pathways utilize and/or build on this concept (15, 38–43).

A recently developed, highly efficient, computational methodology by reference 44 allows for the analysis of a large number of pathway configuration variants and combines the maximization of chemiosmotic energy yield, earlier explored in reference 45, with the maximization of the MDF. The methodology formulates the energetic constrains within the pathway into a mixed-integer linear programming (MILP) problem that can be readily solved using the epsilon-constraint approach (46). Through this methodology, the exploration of the thermodynamic landscape quantifying the trade-offs between energy yield and rate for large numbers of metabolic pathway configurations is now within computational reach. It also constitutes an advancement in our ability to understand and optimize metabolic processes, as it provides a means of identifying the most

**TABLE 1** Main characteristics of the prokaryotic carbon fixation pathways analyzed[a]

| Pathway | Oxygen sensitivity | End product | ATP cost via SLP ($mol_{ATP}$/ $mol_{product}$) | Number of variants | Number of elementary reactions |
|---|---|---|---|---|---|
| Reverse TCA cycle | Anaerobic | Acetyl-CoA | 2 | 54 | 11 |
| Dicarboxylate hydroxybutyrate cycle | Anaerobic | Acetyl-CoA | 3.5 | 216 | 16 |
| Hydroxypropionate hydroxybutyrate cycle | Aerobic | Acetyl-CoA | 5 | 32 | 17 |
| Hydroxypropionate bicycle | Aerobic | Pyruvate | 6 | 12 | 22 |
| Methanogenic Wood-Ljungdahl pathway | Anaerobic | Acetyl-CoA | 0 | 1 | 9 |
| Acetogenic Wood-Ljungdahl pathway | Anaerobic | Acetyl-CoA | 1 | 12 | 9 |

[a]Note that the number of elementary reactions does not include the electron carrier regeneration reactions. The number of such steps varies from four to six depending on the variant. SLP, substrate-level phosphorylation.

efficient pathway configurations for specific environmental conditions or biotechnological applications.

In this study, a comprehensive analysis of both existing (reported) and postulated (hypothetical candidates for pathway engineering) prokaryotic autotrophic carbon fixation pathways is presented to determine those configurations that could yield superior candidates for the engineering of strains that maintain fitness in mixed culture process environments.

The investigation, utilizing both thermodynamic and kinetic principles, takes into account potential variations in these pathways in terms of electron carriers used and energy recovery sites, with a focus on the impact of environmental factors such as $CO_2$ concentration and electron donor availability on the energy costs and driving forces involved in these processes. Additionally, the study aims to determine the optimum electron carrier potentials, with the goal of providing a first-principles explanation for the evolution and selection of specific electron carriers in particular pathway steps. By integrating these various factors and analyzing their interactions, this work offers a comprehensive understanding of the underlying principles governing the efficiency and evolution of these essential carbon fixation pathways in prokaryotes, as well as elucidating the bioenergetic advantages or shortcomings of postulated pathways to be implemented for bottom-up approaches of microbial engineering.

## RESULTS

### ATP cost as function of environmental concentrations

The first analysis is that of the impact of the concentration of carbon dioxide and of the electron donor (hydrogen) on the metabolic energy cost of $CO_2$ fixation (driving forces are not considered here). The net ATP costs (expressed per $CO_2$ molecule fixed) are presented as contour plots in Fig. 1 after each of the six pathways is optimized, noting down the net ATP cost of the most efficient variant over the range of [$H_2$] and [$CO_2$] values.

The results indicate that the way that different concentrations of $CO_2$ and $H_2$ impact the ATP cost is very different among the pathways. The rTCA cycle has the lowest ATP cost per $CO_2$ across all conditions, followed closely by both the Wood-Ljungdahl (WL) pathways. The results also reveal that the anaerobic pathways are more efficient energy-wise, i.e., requiring a lower ATP cost per carbon fixed, compared to the aerobic ones under similar environmental conditions. Prior work suggested that the generally lower reduction potential of anaerobic environments is the main reason why anaerobic carbon fixation is more efficient (15); our results provide an alternative mechanistic explanation to this phenomena.

The rTCA cycle displays contour lines that are all parallel and equally spaced, indicating that the ATP cost changes almost logarithmically with respect to the $CO_2$ and $H_2$ concentrations as it does the theoretical Gibbs energy. In contrast, the 3HP-4HB cycle and the 3HP bicycle, typically present in aerobic organisms, display contour lines that are spaced far apart, indicating ATP costs for carbon fixation are largely unaffected for a wide range (several orders of magnitude) of concentration values. In addition, the ATP costs of carbon fixation for both the pathways are the highest among all the pathways. Since aerobic catabolisms are energy rich in ATP yields, less efficient anabolic pathways are justified in exchange for higher driving forces and rates. It is therefore very unlikely that a microbe will operate an aerobic carbon fixation pathway under a yield over rate strategy.

The WL pathways exhibit irregular contour lines in ATP cost for the different $CO_2$ and $H_2$ concentrations. Neither the acetogenic nor the methanogenic pathway appears more efficient than the other in general, except in possible small concentration regions in which one or the other could fix $CO_2$ at a slightly smaller cost.

The analysis was repeated in the same concentration ranges but imposing now an MDF of −5.45 kJ/mol (equivalent to 80% FFE) to all reaction steps. The resulting contour plots are presented in Fig. 2 in which most of the patterns described under no imposed MDF still are observed. However, the aerobic (3HP-4HB cycle and 3HP bicycle) pathways

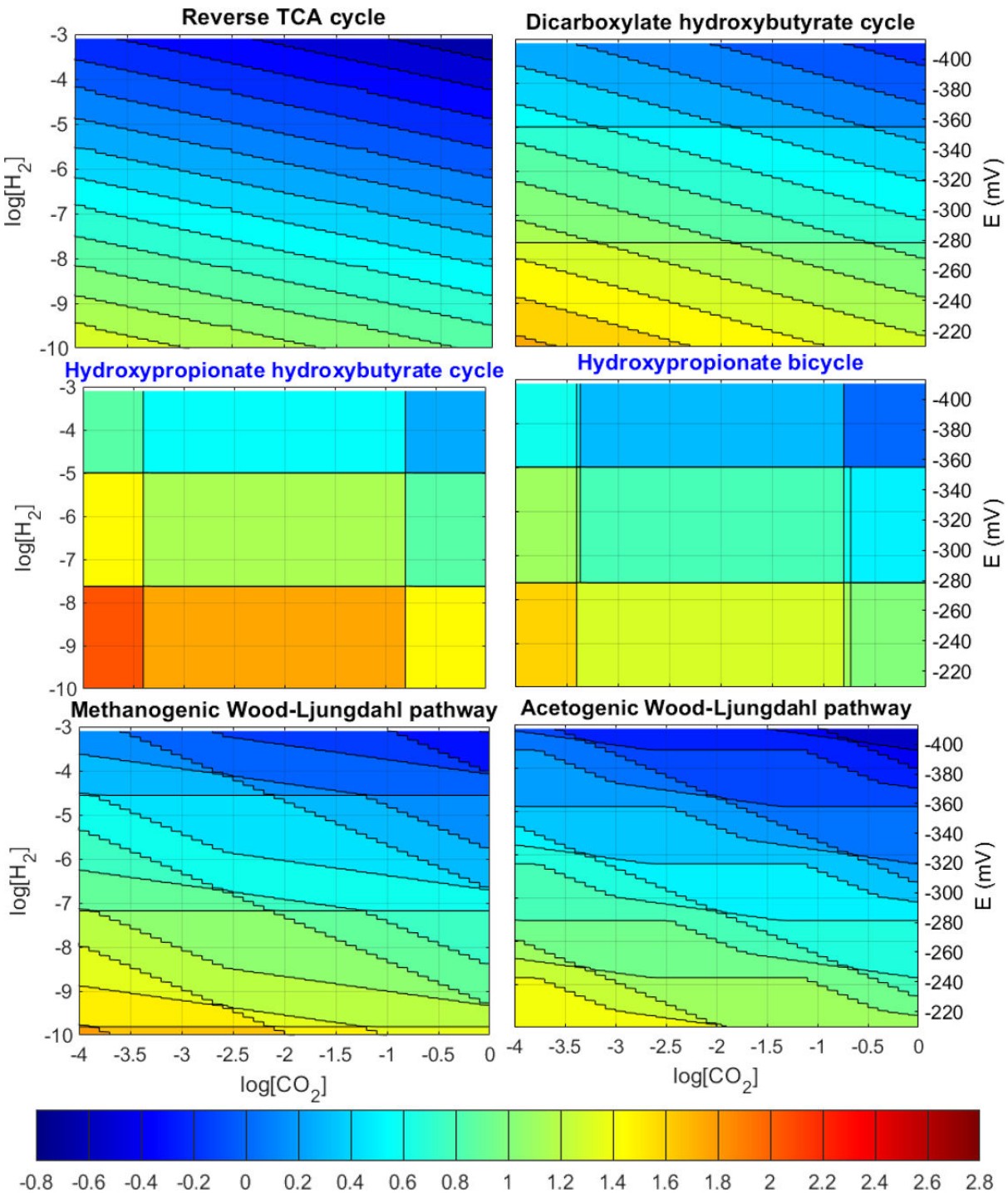

**FIG 1** Contour plot of ATP cost for carbon fixation in $mol_{ATP}/mol_{CO2}$ for each of the six major microbial pathways with no minimum driving force imposed. The right *y*-axis indicates an equivalent electron donor reduction potential to the hydrogen concentration. Negative ATP cost values indicate that net ATP is produced. The middle two pathways in blue font are aerobic, all other anaerobic.

under imposed MDF now appear capable of recovering energy more efficiently while maintaining large driving forces, as shown by the larger number of contour lines and smaller regions of constant yield.

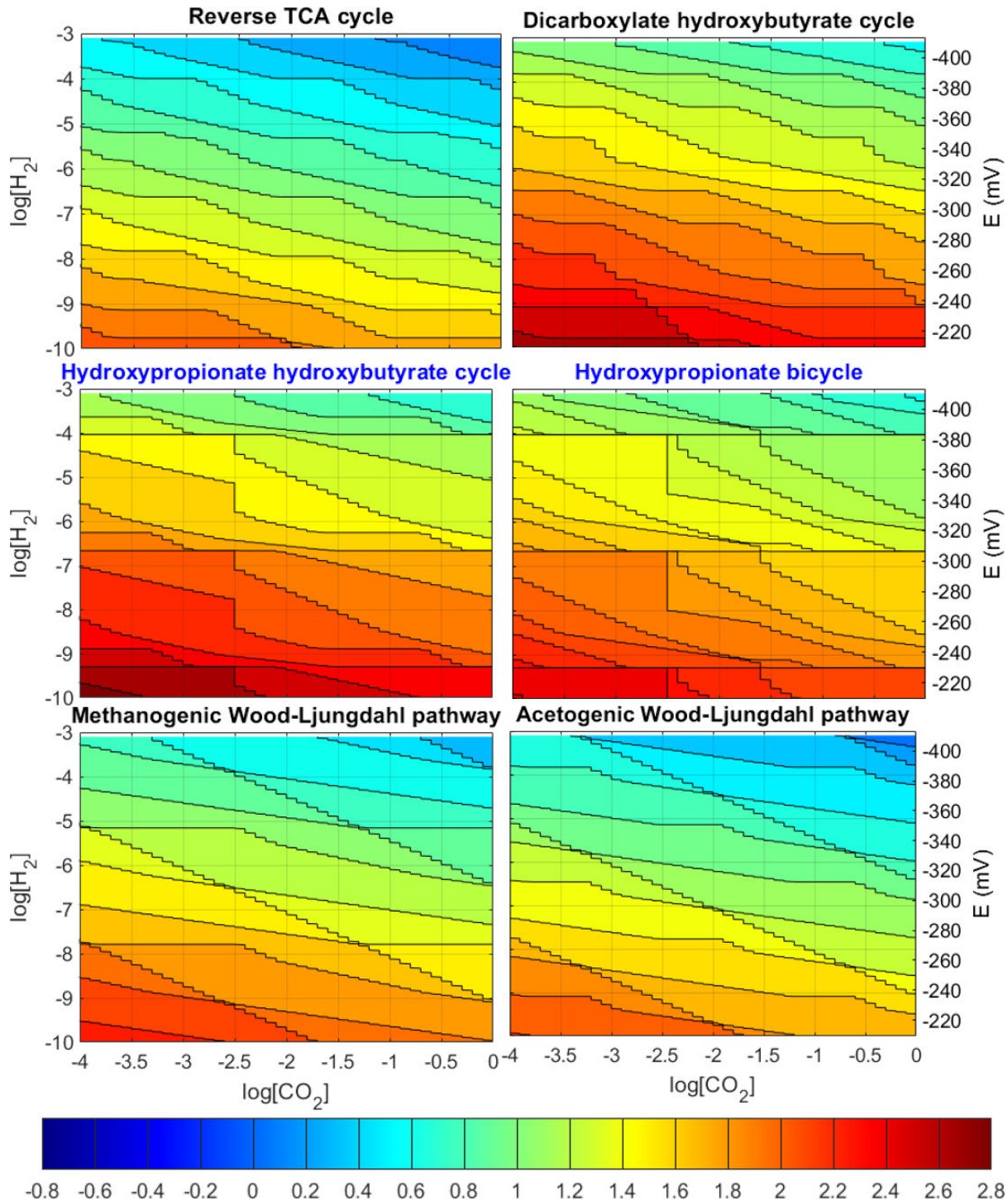

**FIG 2** Contour plot of ATP cost for carbon fixation in $mol_{ATP}/mol_{CO_2}$ for each of the six major microbial pathways with an MDF of 5.45 kJ/mol is imposed. The right *y*-axis indicates an equivalent electron donor reduction potential to the hydrogen concentration. Negative ATP cost values indicate that net ATP is produced. The middle two pathways in blue font are aerobic, all other anaerobic.

## Trade-offs between energy cost and MDF

To study the trade-off between energy cost and driving force, all six pathways were optimized (as described in Materials and Methods), sequentially for energy yield and MDF. Figure 3 shows the optimum net ATP costs per $CO_2$ fixed and the maximum MDF

achieved for all pathways as a function of the concentration of the electron donor (hydrogen). This is shown for the cases of none, one, two and three chemiosmotic proton translocations (CPTs) traded in exchange for driving force.

The periodic behavior displayed by the MDF curves as a function of the $[H_2]$ is expected, since the larger the $[H_2]$, the larger the overall available Gibbs energy and driving force, until the threshold for the recovery of one additional CPT is reached. That energy is therefore no longer available as driving force but as ATP and the MDF drops instantaneously accordingly.

When at the threshold points, the most efficient (zero protons traded) MDF curve falls close to nearly zero; this indicates that there exists a set of feasible intermediate concentrations that lead to an almost 100% recovery of the available Gibbs energy, which makes all the reaction steps very close to the thermodynamic equilibrium. While this is a scenario of no practical use for the cell as the enzymatic extremely low FFE would stop the pathway, this shows the upper limit of the yield vs driving force trade-off space from which there is flexibility to trade yield for driving force.

The MDF and energy cost transition patterns and intervals vary between pathways and are not always repeating predictably within the same pathway. This is a result of the non-linearities in the optimal net ATP costs that were discussed in the previous subsection. The amount of extra driving force (MDF) generated can vary significantly between the initial and subsequent CPT trade-offs for certain pathways. Specifically, the 3HP-4HB cycle and the 3HP bicycle exhibit the capability to generate large thermodynamic driving forces while incurring only moderate ATP yield losses as seen in Fig. 3. This is true for all cases with and without traded CPTs. Without CPTs traded, the 3HP-4HB cycle can achieve an MDF over 4 kJ/mol, and the 3HP bicycle can achieve values over 3 kJ/mol. This is where these aerobic pathways show great advantage; with the right set of conditions, they can achieve remarkably high MDF values that could be translated into fast kinetics without increasing the ATP costs. On the other hand, the rTCA cycle performs quite poorly in the sense of "peak" MDF reaching only an MDF value of around 1 kJ/mol without any CPT trade, leading a much lower FFE.

The figure also reveals the ability of the different pathways to utilize the additional energy available from traded CPTs. For example, the MDF curves of the 3HP-4HB and

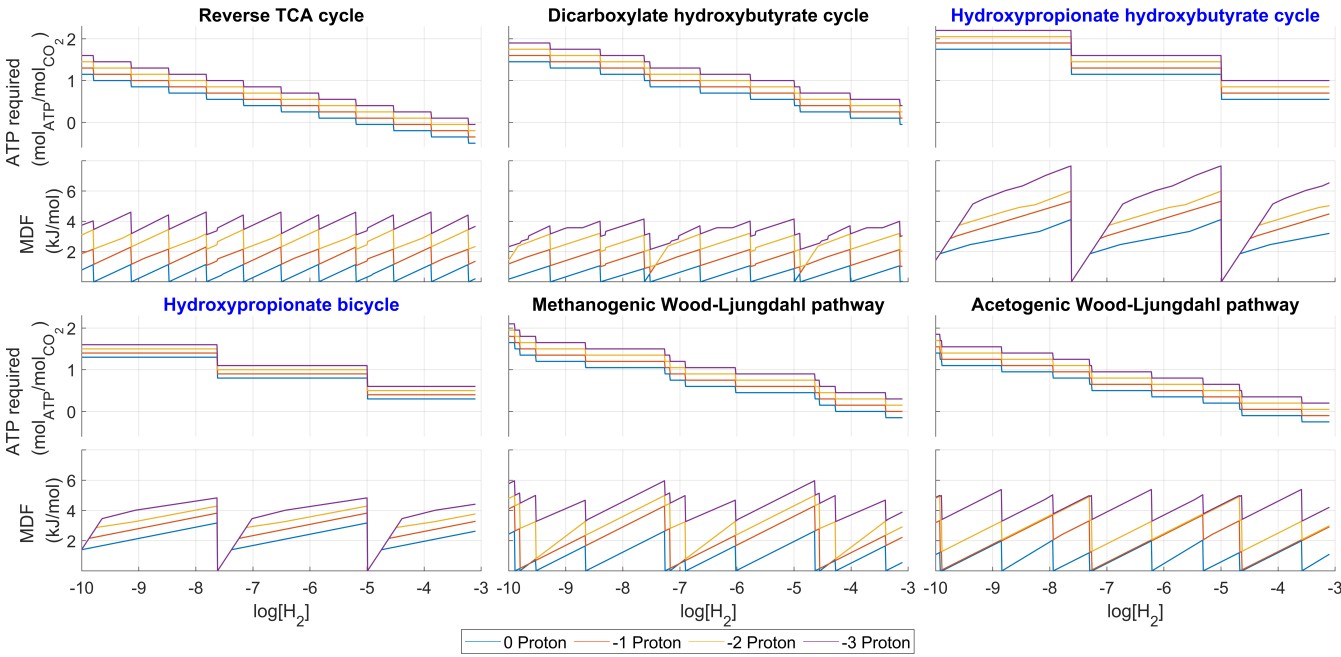

**FIG 3** ATP costs and MDF values of all six pathways function of the $H_2$ concentrations ($[CO_2]$ set at 10 mM). The different lines correspond to zero (blue), one (green), two (cyan), and three (red) chemiosmotic proton translocations traded in exchange for driving force.

3HP bicycle pathways always drop back to nearly zero at their troughs, overlapping completely for a short interval of $[H_2]$ at the beginning of every cycle. This means that for these two pathways, no MDF benefit is gained in those intervals from the traded CPTs for increased ATP cost. One possible cause for this is the presence of a bottleneck reaction that is at a local maximum of driving force, preventing any further increase in the pathway MDF without significant pathway disruption. In those conditions, it is not worth it for the cell to try to increase the driving forces by trading energy. However, for the rTCA and DCHB cycles, each MDF curve remains always above the curves with less protons traded. This implies that these pathways always effectively translate all energy of the traded CPTs (i.e., additional energy costs) directly into higher MDF values.

In general, it is observed that the gain in MDF is not constant with respect to the number of protons traded in all but the rTCA cycle pathway. The question naturally arises as to in what cases a large loss of MDF is justified to achieve a small increase in yield.

## Bioenergetics trade-offs for postulated carbon fixation pathways

In an equivalent manner to the natural pathways, a series of postulated pathways were also evaluated. These are hypothetical and assembled from existing prokaryotic enzymatic reactions as candidates for possible synthetic pathway engineering. The impact of $CO_2$ and $H_2$ concentrations on their net ATP costs and driving forces was evaluated. The results are shown in Fig. 4 and 5, respectively.

The contour plots of ATP costs per carbon fixed are shown in Fig. 4. The shortest pathways (via citrate and via pyruvate) show patterns similar to those of the rTCA cycle. The high density of contour lines signifies the flexibility of these pathways in terms of ATP cost; these pathways are able to capture additional energy in the electron donor or $CO_2$ in the form of a CPT with relative ease. On the other hand, the via malonyl-CoA pathways show similar behavior to the 3HP-4HB cycle and 3HP bicycle, with their contour lines remaining flat across orders of magnitude of $CO_2$ and $H_2$ concentrations. This indicates that the via malonyl-CoA postulated pathways are not optimized for substrate utilization efficiency. For the CETCH cycle pathways, the complicated pattern of ATP cost indicates that different reactions steps could be acting as bottlenecks at different conditions, requiring a more detailed analysis of the pathway steps similar to that shown in Fig. S3.

It is interesting to note that pathways which assimilate glyoxylate via the glyoxylate shunt require a lower net ATP cost than those utilizing the tartronate semialdehyde reaction, despite the former being significantly longer than the latter. This difference can be partially explained by the difference in the end product: the glycerate from tartronate semialdehyde assimilation route is 2/3 ATPs richer in energy than the pyruvate from glyoxylate shunt (assuming the conversion between the two using the glycolysis pathway). Even so, the difference in energy still slightly favors the glyoxylate shunt route, indicating that it can be more optimized for energy recovery efficiency.

The trade-off between energy cost and driving force for the postulated pathways is shown in Fig. 5 for a range of values of $\log[H_2]$. The graphs show periodic triangular plots for the same reasons as the natural pathways. However, the periods are much less regular in the postulated pathways, again suggesting that different reactions may be acting as bottlenecks at different conditions (values of $\log[H_2]$).

Unlike the naturally occurring aerobic pathways, the via malonyl-CoA pathways do not attain large values of MDF at optimal yield (no protons traded). In fact, a plateau occurs on the MDF curves for all the postulated pathways with the exception of the via pyruvate and via citrate routes. This suggests that a specific bottleneck reaction cannot attain any higher driving forces, and the additional Gibbs energy cannot be moved to any chemiosmotic energy recovery site. Upon closer internal inspection of the pathways, the bottleneck appears always in the reaction where the final three-carbon product is formed with the release of a $CO_2$ molecule. The occurrence of $CO_2$ as a pathway product in addition to being a substrate at the carboxylation steps of the pathways constrains the cell to work at moderate $CO_2$ concentrations rather than attempting to maximize it to

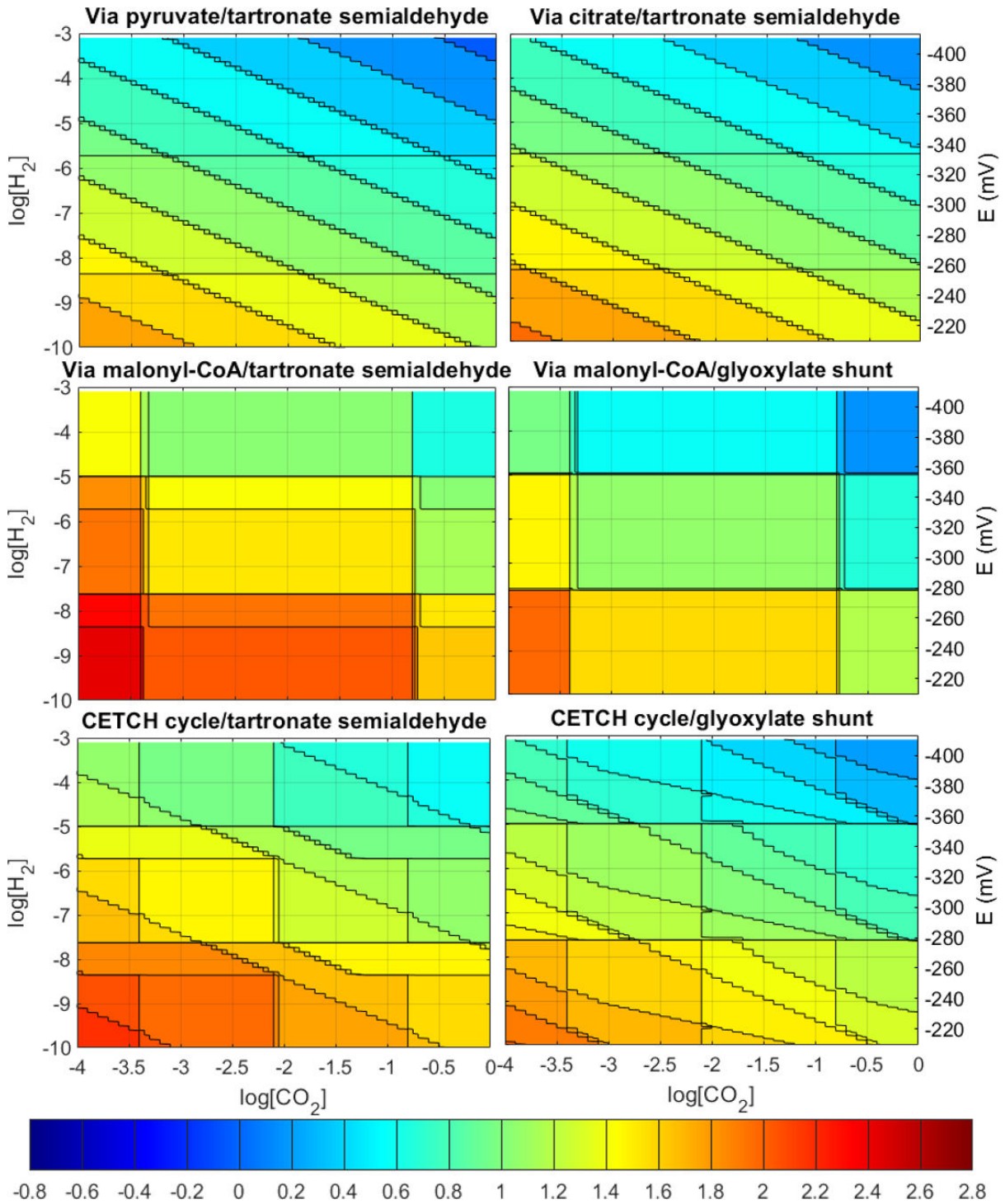

**FIG 4** Contour plot of ATP cost for carbon fixation in $mol_{ATP}/mol_{CO2}$ for the postulated pathways with no MDF imposed. The right *y*-axis indicates the equivalent reduction potential to the hydrogen concentration. Negative ATP cost values indicate that net ATP is produced.

drive the thermodynamics of carboxylation reactions. This severely restricts the bioenergetics of the individual pathway reaction steps and could partially explain why, despite the pathways being overall more energetically favorable, these pathways have not been observed in any naturally occurring autotrophs.

This bottleneck is so severe that even with the concession of multiple protons, the MDF of the pathways does not increase. In fact, at those plateaus, the concentration of

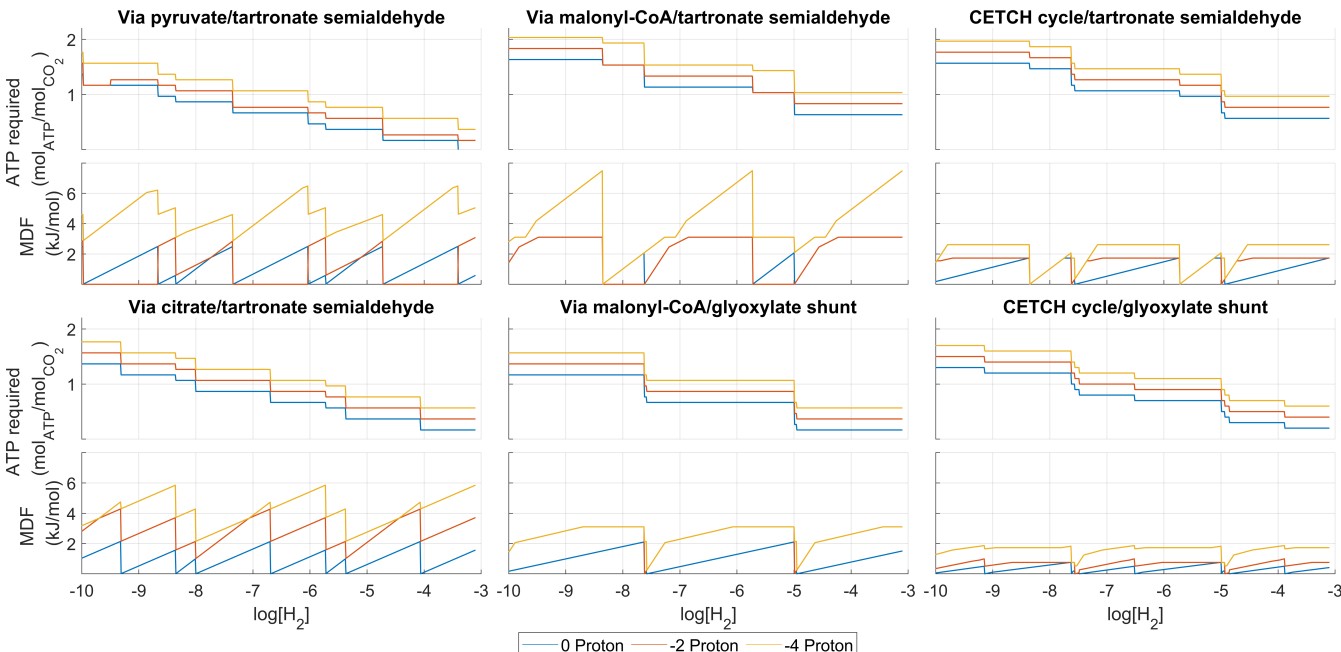

**FIG 5** Optimized ATP costs and MDF values for postulated pathways at different electron donor (hydrogen) concentrations. The $CO_2$ concentration is set at 10 mM. The optimal lines imply no CPTs traded. The other lines are obtained by changing the value of ε (number of CPTs traded).

the substrate for the bottleneck reaction (tartronate semialdehyde/malate) is at the maximum permissible limit and the concentration of the organic product (glycerate/pyruvate) is at the lower limit. The only way the driving force could be further increased at that point is if the electron carrier performing the reductive decarboxylation was more energetic (higher ratio of $[eC]_{red}/[eC]_{ox}$). While we assume that this ratio is flexible and subject to optimization in this work, this is not the case in many microbes as these carriers participate in hundreds of reactions across the entire metabolism of the cell, making it impossible to increase the MDF of these pathways any further under realistic conditions.

## Seeking for the optimal electron carrier potentials from first principles

An additional objective of this study was to determine what the optimum potential for each electron carrier is and to provide a first-principles explanation for the biochemical selection among the possible electron carriers in specific pathway steps. For this purpose, all six selected natural pathways were evaluated again but with all redox reactions described such that the electrons are accounted for by their potentials alone (i.e., without any biochemical electron carrier defined). All possible solution pools were evaluated to reveal the ranges of electron carrier potentials at each reaction step that maintain ATP cost and MDF at the optimum, and this is then evaluated for different $H_2$ concentrations. Under this approach, upper and lower boundaries of permitted electron carrier potential values are generated. In Fig. 6, the results for the rTCA and DCHB cycle redox reaction steps are displayed. Complete results for the other pathways are presented in the Supplemental Material section S4.

In the results, a diversity of patterns for the different redox reaction steps can be observed. Some reactions show very narrow ranges of optimal potentials (<100 mV), such as the reduction of succinyl-CoA in the rTCA cycle, as well as the oxidation of succinate semialdehyde and reduction of 3-hydroxybutyryl-CoA in the DCHB cycle. Other reactions such as the reduction of KGA in the rTCA cycle and the reduction of OAA in both pathways show a wide range of optimal potentials (>300 mV). Reaction steps that have a broader optimum potential range for the electron transfer will overlap with more existing biochemically known electron carriers that could therefore be optimally selected

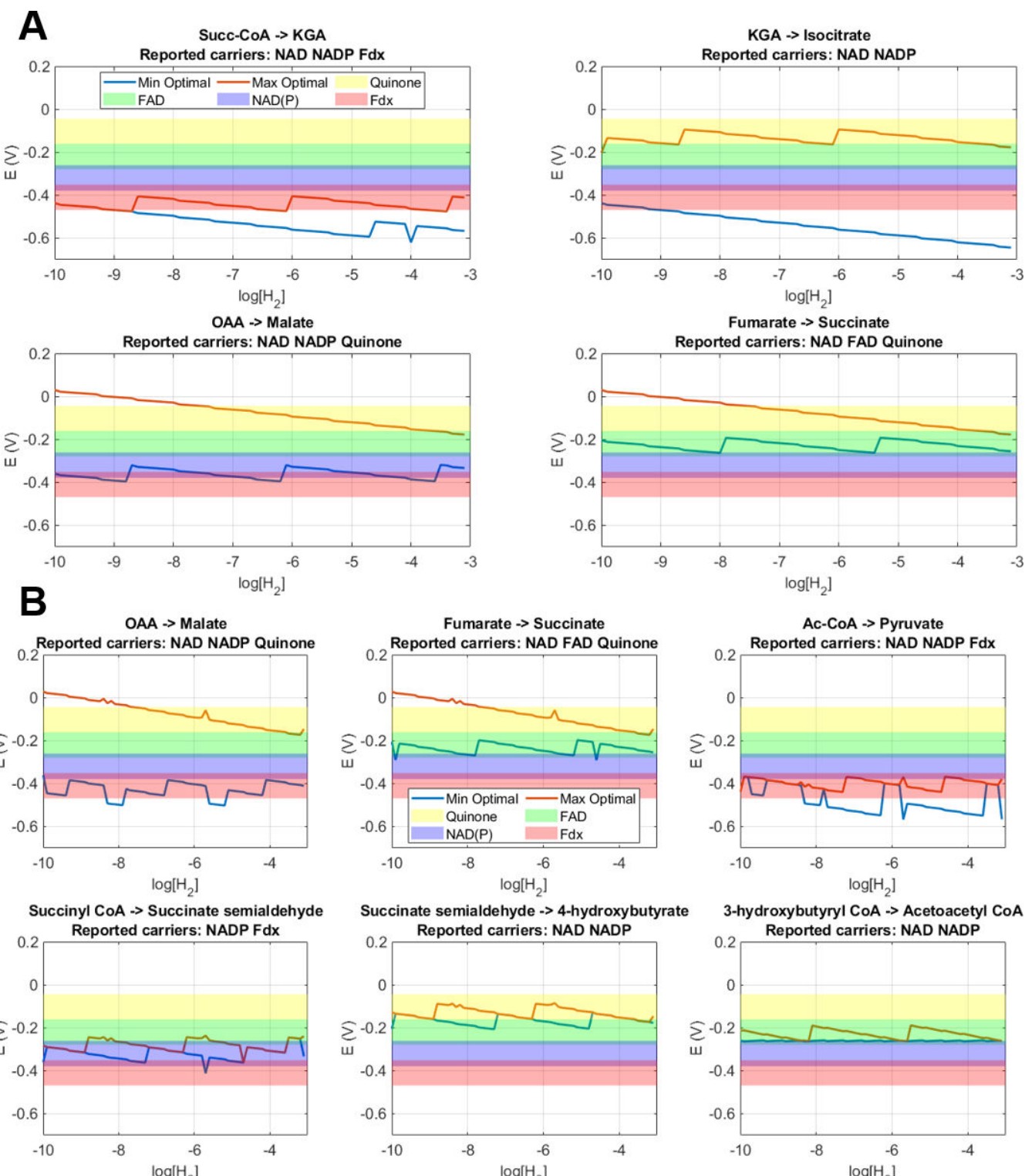

**FIG 6** Optimal electron carrier potential ranges for the redox steps in the rTCA (A) and DCHB (B) cycle pathways as a function of the electron donor concentration. The shaded ranges represent the potentials of biochemically known electron carriers (EC) across a concentration ratio ([$EC_{red}$]/[$EC_{ox}$]) range from $10^{-2}$ to $10^{2}$. The two lines represent the region of optimal (i.e., resulting in the lowest pathway ATP cost as well as maximum MDF) EC potential as a function of [$H_2$]. These lines are obtained by first letting the optimization solver find the optimal pathway ATP cost and MDF with no restriction on the EC reduction potentials, then setting these optimal values obtained as additional equality constraints and sequentially minimizing and maximizing the reduction potential of each EC involved in the pathway.

to run that redox reaction step. Those reactions with narrower potential ranges may be restricted to only one specific electron carrier if to remain optimal. The broader ranges appear to occur in reaction steps that can also translocate protons, thus allowing the cell more freedom to draw/supply energy for the redox reaction from the chemiosmotic proton pool rather than the electron carrier itself.

## DISCUSSION

A comprehensive optimization methodology, applied to known prokaryotic autotrophic $CO_2$ fixation pathways, allowed for the determination of their optimal configurations in terms of ATP yield as a function of environmental conditions. Additionally, the methodology was used to quantify the trade-offs if the cells were to trade energy efficiency (ATP yield) for driving force (dissipated energy), potentially allowing for higher rates.

The pathways known to occur in anaerobic organisms can fix $CO_2$ at a lower net ATP cost compared to those found in aerobic organisms, spanning a broad range of $CO_2$ and electron donor ($H_2$) concentrations. The reverse TCA cycle pathway exhibits the lowest overall ATP cost and maximum flexibility, even across diverse $CO_2$ and $H_2$ conditions, thanks to its multiple possible proton-translocating sites. Consistently, the reverse tricarboxylic acid cycle and the Wood-Ljungdahl pathways display high energy efficiency under a broad range of conditions.

Pathways with many contour lines in Fig. 1 and 4 indicate that the ATP cost is highly sensitive to changes in the environmental concentrations, as compared to pathways with less contour lines. This reflects the ability of those pathways to recover more ATP due to more chemiosmotic energy recovery sites in the pathway; for example, the rTCA cycle, which has the largest number of contour lines, has three such chemiosmotic energy recovery reaction steps (namely the reductions of KGA, OAA, and fumarate), the most out of all the pathways. In a real ecosystem, this could be translated into an efficient use of resources for microbes with changing environments, or a better capability to recover energy from even small changes in environmental conditions.

Although the number of SLP (substrate level phosphorylation) steps is very different between pathways (from none in the methanogenic WL to six in the 3HP bicycle), our results indicate that pathways have roughly the same ATP costs for carbon fixation after accounting for CPTs. The difference between the most and the least energetically efficient pathways is small, ranging between 0.5 and 1 $mol_{ATP}/mol_{CO2}$, depending on the environmental concentrations. This indicates that these pathways can become much more optimized at the chemiosmotic level beyond their well-described SLP ATP yields.

The pathway analysis also reveals that the trade-off relationship between energy yield and driving force is not constant (the gain in driving force for each traded CPT changes depending on the environmental conditions) and non-linear (additional CPT concessions yield varying amounts of extra driving force). This can be construed from the changing slopes of some of the MDF curves in Fig. 3, particularly those of the DCHB, 3HP-4HB, and 3HP (bi)cycles. As the slope represents the gain in MDF for a small increase in $C_{H2}$, a change in slope indicates that a different reaction is now limiting the optimum MDF for that pathway. Identification of the specific reaction bottleneck could be possible by imposing a slightly higher MDF constraint on the pathway, then using the mathematical concept of irreducible inconsistent subsets (a mathematical optimization technique that produces a complete and sufficient set of active constraints [47]). In addition, further insights for any specific condition can be obtained by examining the optimal intermediate metabolite concentrations of any pathway (e.g., as in Fig. S3).

The methodology was also applied to the analysis of selected hypothetical postulated pathways assembled from existing biochemistry literature as candidates for possible pathway engineering. The postulated pathways studied show generally higher ATP costs than the natural pathways and are not capable of building up higher MDF values as a trade-off for the above. This could partially explain why these pathways have not yet been identified in any naturally occurring autotroph. There is a need for the search

for other more efficient hypothetical pathways to continue by the wider microbial engineering community, as these pathways are theoretically not more effective than the natural pathways.

In this study, the optimum potentials for each electron carrier were also investigated to provide a first-principles explanation for the biochemical selection of electron carriers in specific pathway steps. Our results showing electron carrier optimal potential ranges shed light on another avenue of thermodynamic adaptation in prokaryotic biochemistry. Interestingly, the optimal potential range for a given redox reaction step appears to not change, irrespective of in what pathway the reaction step appears. In additional simulations (not shown) seeking higher MDF by trading three CPTs, the optimal potential ranges showed very little change. This could suggest that one specific optimal electron carrier (at a given $[eC_{red}]/[eC_{ox}]$ ratio) may exist for the particular reaction irrespective of the cell strategy toward high efficiency (yield) or high driving force (rate).

Comparison of the optimal potential ranges predicted with the known biochemical electron carriers confirms that many reactions indeed utilize the optimum electron carriers. As an example, the reduction of fumarate to succinate appears to be optimal with quinones at lower and with FAD at higher hydrogen concentrations. Both these carriers have indeed been reported as the ones used for this reaction (48, 49) and the third experimentally confirmed carrier, NAD, is also in the optimal range but only in high (above $10^2$) [NADH]/[NAD] ratios. In other cases, such as for the oxidation of succinate semialdehyde where quinone would appear as the optimum carrier, the reported enzymes use either NAD or NADP, which suggest that specific biochemical constraints are at play or that the organisms using the optimum carrier (quinone) are yet to be reported. These specific cases suggest hypotheses for potential pathway enhancement and possible targets for metabolic pathway engineering. These results provide for the first time a mechanistic interpretation that partly confirms the choices of electron carriers from first principles while identifying potential sites for targeted pathway enhancement or engineering.

The development of cost-effective microbial $CO_2$ fixation processes using existing or engineered microbial strains requires a better understanding of the pathway capabilities under different operation conditions, and of their bottlenecks including the trade-offs between energy conservation and reaction rate. The above results bring about mechanistic insights and empower researchers to intelligently operate or direct strain engineering efforts among the potentially vast landscape of options to follow toward industrial mixed culture growth.

## MATERIALS AND METHODS

The relevant information about the six prokaryotic autotrophic carbon fixing metabolic pathways evaluated is presented, followed by a summary of the modeling approach used to formulate the problem into an MILP and subsequently solve it.

### Selection of pathways and their configuration variants

#### Naturally occurring pathways

Six pathways for prokaryotic carbon fixation were identified and selected from existing literature. The balanced overall equations, when hydrogen is used as electron donor, for all the pathways (except the 3HP bicycle), are given as follows:

$$2CO_2 + 4H_2 + CoA \rightarrow AcCoA + 3H_2O \qquad \Delta G^0 = -142.3 \text{ kJ/mol}$$

and for 3HP bicycle:

$$3CO_2 + 5H_2 \rightarrow \text{pyruvate} + 4H_2O + H^+ \qquad \Delta G^0 = -113.6 \text{ kJ/mol}$$

Our analysis will consider dissolved hydrogen as electron donor; however, alternative energetically equivalent donors could be used instead, including bioelectrochemical direct electron transfer at equivalent voltages.

The complete description of the pathways in terms of metabolites, chemiosmotic energy conservation sites, and permissible electron carriers was assembled from biological databases such as MetaCyc and the Kyoto Encyclopedia of Genes and Genomes (50, 51), and other literature sources (15, 27, 35, 37, 52). These complete details are presented in full in the Supplementary Material section S2. Summaries of the most relevant biochemical details and individual reaction steps for the six pathways are shown in Table 1 and Fig. 7, respectively. All reactions in the pathways are considered to take place within a single cell/compartment, and the final products as remaining within the cell (anabolic carbon fixation). For each of the pathways, biochemically feasible configuration variants were derived based on the possible electron carrier combination choices, see references 44, 45.

## Synthetic pathways for $CO_2$ fixation

In addition to the natural prokaryotic pathways for $CO_2$ fixation listed in the previous subsections, many researchers have proposed alternative pathways for the conversion of inorganic $CO_2$ into central metabolites. These postulated pathways are based on combinations of real existing prokaryotic enzymes that lead to shorter or more stable cycles. In this work, six such novel metabolic pathways are considered for thermodynamic analysis.

One common feature for all these pathways is that their end product is glyoxylate, a two-carbon metabolite that requires further biochemical transformation to arrive to the central metabolites considered in the previous subsection (either acetyl-CoA or pyruvate). Two possible routes have been considered for this conversion: the glyoxylate shunt pathway which converts two glyoxylate molecules into pyruvate, and the tartronate semialdehyde pathway which converts two glyoxylate molecules into glycerate (52). Both routes involve the loss of one carbon atom as $CO_2$, leading to an overall carbon atom economy for the fixation process of 75%.

Summaries of the most relevant biochemical details and individual reaction steps for the six synthetic pathways are shown in Table 2 and Fig. 8, respectively.

## Methodology for pathway trade-offs analysis

The bioenergetic evaluation of the possible pathway variants is conducted as per the methodology described in reference 44. The basis of the approach is based on the assumption that microbes performing near or at the optimal efficient use of resources (ATP energy yield) and fastest metabolic growth rates (avoiding small driving forces) have an advantage and have been and will be positively selected in a given environment or bioreaction process. Since these objectives are generally conflicting, depending on the environmental conditions, an optimum trade-off exists. In order to be able to evaluate these trade-offs, the methodology formulates a multi-objective optimization problem with the following two objectives:

$$\max_{\underline{x}} \underline{J} = \begin{bmatrix} \sum n_{p,j} \\ B \end{bmatrix} \qquad (2)$$

subject to

$$\underline{\underline{S}} * \begin{bmatrix} \ln C_1 \\ \ln C_2 \\ \vdots \\ \ln C_n \end{bmatrix} + \text{pmf} * \begin{bmatrix} n_{p,1} \\ n_{p,2} \\ \vdots \\ n_{p,j} \end{bmatrix} + \begin{bmatrix} F_1 \\ F_2 \\ \vdots \\ F_j \end{bmatrix} \leq \begin{bmatrix} \ln K_1 \\ \ln K_2 \\ \vdots \\ \ln K_j \end{bmatrix}, \qquad n_{p,j} \in Z \qquad (3)$$

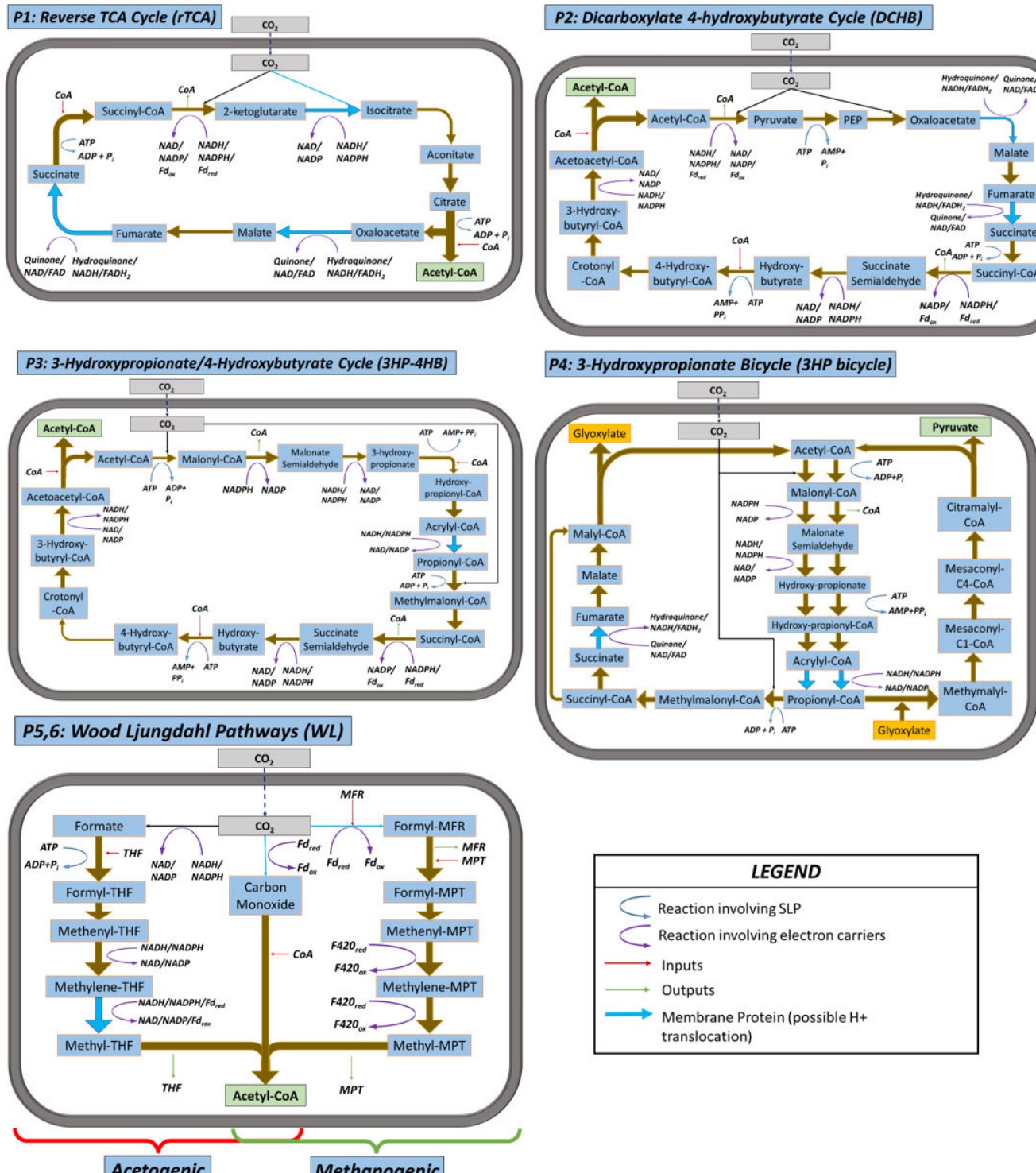

**FIG 7** Overview of the prokaryotic carbon fixation pathways analyzed.

where $F_j$ is the driving force term for reaction $j$ given by

$$F_j = -\frac{\Delta G_{\min, j}}{RT} \qquad (4)$$

**TABLE 2** Main characteristics of the synthetic carbon fixation pathways analyzed[a]

| Pathway | Glyoxylate assimilation route | End product | ATP cost via SLP $(mol_{ATP}/mol_{product})$ | Number of variants | Number of elementary reactions | Reference |
|---|---|---|---|---|---|---|
| Via pyruvate | Tartronate semialdehyde | Glycerate | 2 | 12 | 14 | (53) |
| Via citrate | Tartronate semialdehyde | Glycerate | 2 | 12 | 14 | (53) |
| Via malonyl-CoA | Tartronate semialdehyde | Glycerate | 7 | 4 | 20 | (54) |
| Via malonyl-CoA | Glyoxylate shunt | Pyruvate | 8 | 36 | 23 | (54) |
| CETCH cycle | Tartronate semialdehyde | Glycerate | 5 | 16 | 20 | (55) |
| CETCH cycle | Glyoxylate shunt | Pyruvate | 6 | 144 | 23 | (55) |

[a]Note that the number of elementary reactions does not include the electron carrier regeneration reactions. The number of such steps varies from four to seven, depending on the variant.

$n_{p,j}$ is the number of protons translocated in reaction $j$, $B$ is the smallest of the driving force terms ($F_j$) among all reactions, $\underline{S}$ is the stoichiometry matrix (representing the pathway reaction network) where an $S_{ij}$ entry is the stoichiometric coefficient of species $i$ in reaction $j$, $C_i$ is the molar concentration of species $i$ (mol/L), $K_j$ is the equilibrium constant for reaction $j$ (incorporating already the energy contribution of any SLP that may occur in that reaction step), and pmf is the energy yielded by one membrane CPT). The decision variable vector consists of finding the optimum logarithmic concentrations of all species involved in the pathway as well as the discrete (integer) number of protons translocations involved in each reaction step in which it is biochemically permitted (i.e., reactions catalyzed by a membrane-bound enzyme capable of chemiosmotic energy conservation). This problem formulation allows for decision variables and objective function to have a linear dependence leading to an optimization problem which is an MILP. Maintaining this linearity allows for the use of very powerful and computationally inexpensive linear optimization methods, which also ensures global optimality of the solutions found and makes it possible to conduct this analysis.

Permissible values of any metabolite concentrations are constrained between $10^{-6}$ and $10^{-2}$ mol/L, based on osmotic and kinetic considerations, in line with other studies (40, 44, 45, 56). The maximum number of CPTs permitted in either direction in a single reaction step is set just under the ATP synthase ratio ($r_{H+/ATP}$), rounded down to the nearest integer.

Figure 9 provides a summary of the pathway analysis procedure. The so-called epsilon-constraint method is used to solve the multi-objective optimization by means of two consecutive runs. In the first run, all driving force terms are set to zero, and the goal is to maximize the sum of CPTs, which leads to the highest energy recovery or efficiency achievable.

If a feasible configuration and solution are found, the second optimization run is executed, where a user-defined number of CPTs is traded (not used) to allow for larger driving forces. This is accomplished by including an additional constraint on the maximum net CPTs into the second optimization run, in the form:

$$\sum n_{p,j} = n_{p,opt} - \varepsilon \qquad (5)$$

where $\varepsilon$ is the user-defined parameter indicating how many (if any) CPTs are allowed to be lost, leading to higher energy dissipation and driving forces. By setting the number of CPTs not used ($\varepsilon$) to a higher number, the more importance is given to driving forces at the expense of energy yield and therefore the trade-off between the two can be quantitatively shown.

The driving force terms ($F_j$) and their smallest value ($B$) are then added to the decision variable vector $\underline{x}$. The objective function in the second run is to maximize $B$, thereby maximizing the smallest of the driving forces $F_j$ among all reaction steps in the pathway variant.

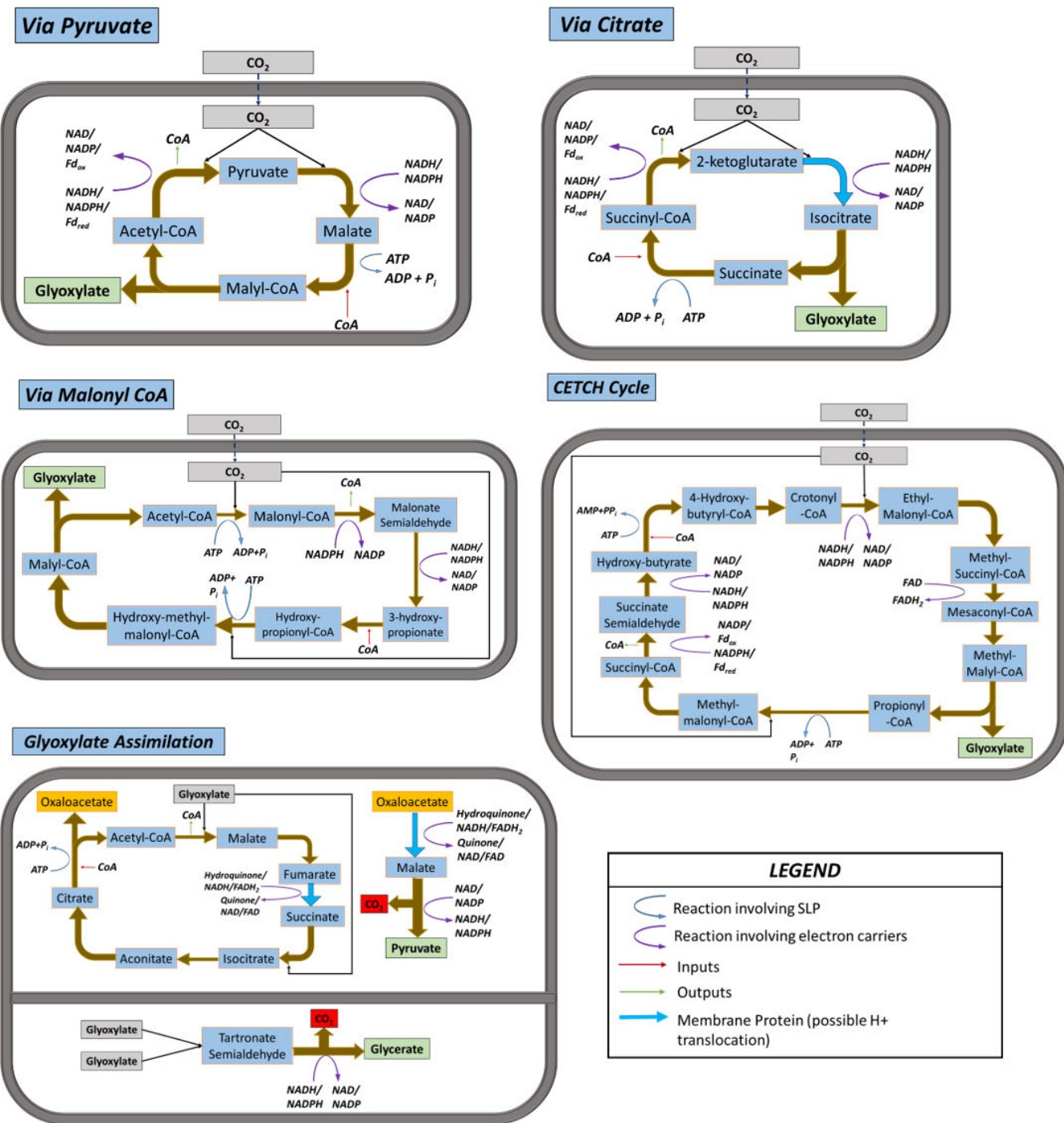

**FIG 8** Overview of the postulated carbon fixation pathways analyzed.

The model was implemented on a cross-platform Excel-MATLAB toolbox. The pathway and species data are stored in a preformatted Excel spreadsheet. The data are then loaded into MATLAB, which subsequently calls on a suitable MILP solver to optimize the pathway. The solver of choice in this work was Gurobi, a specialized optimization toolbox that is considered state-of-the-art in the field of operations research (57).

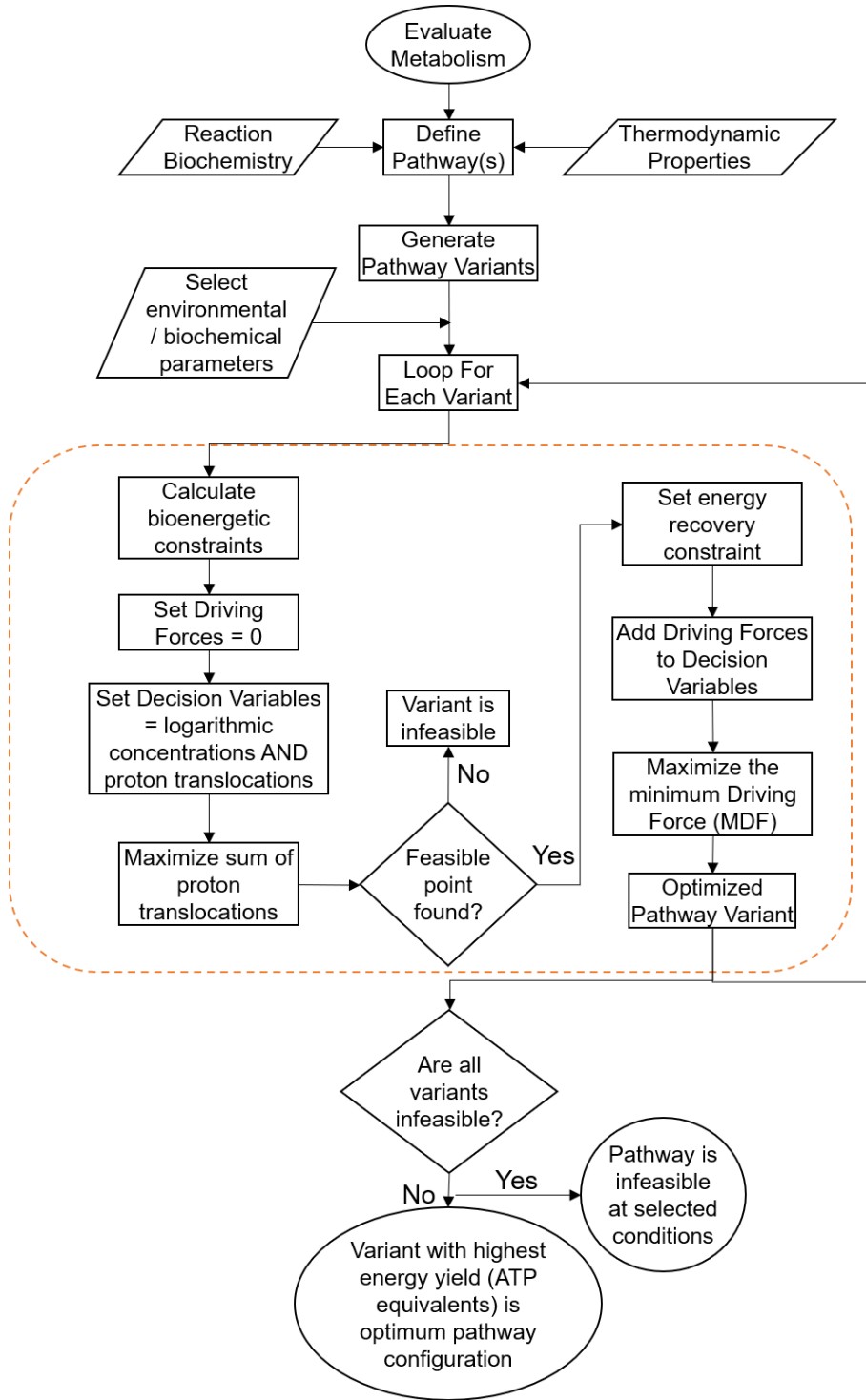

**FIG 9** Flowchart describing the steps involved in the evaluation of a given metabolic pathway variant. The block within the orange dashed box constitutes the steps of one multi-objective optimization run. Adapted from reference 44.

## Optimal electron carrier potentials

A second objective of this study is to determine the optimum potentials at which electrons are to be exchanged with the selected carriers in each reaction step. This aims at finding a first principles-based optimal selection of electron carriers in each particular pathway step. This can not only inform targeted metabolic engineering strategies but

can also bring about insights on the nature of the most common electron carriers as they evolved to deliver electrons at the specific potentials they do. The goal defined was to discover whether it is possible to derive the choice of electron carriers found in carbon fixation pathways from first principles based on the assumption of optimality of microbes for energy yield. To achieve this, the methodology presented in the previous subsection was modified.

To analyze in terms of the electron potentials, all redox reactions in the metabolic pathways were described in terms of their half-equations with the electrons explicitly accounted for by their potentials (i.e., without any biochemical electron carrier defined). Generic unspecified carrier regeneration reactions against the terminal electron donor (hydrogen in this study) were added to ensure that the overall metabolic transformation was closed:

$$2H^+ + 2e^- \rightarrow H_2$$

The energy constraint for the reaction network, after adding a Nernst potential term and rewriting in matrix form, is then given by:

$$\boldsymbol{S} * \begin{bmatrix} \ln C_1 \\ \ln C_2 \\ \vdots \\ \ln C_n \\ \Phi_1 \\ \vdots \\ \Phi_j \end{bmatrix} + \text{pmf} * \begin{bmatrix} n_{p,1} \\ n_{p,2} \\ \vdots \\ n_{p,j} \end{bmatrix} + \begin{bmatrix} F_1 \\ F_2 \\ \vdots \\ F_j \end{bmatrix} \leq \begin{bmatrix} \ln K_1 \\ \ln K_2 \\ \vdots \\ \ln K_j \end{bmatrix}, \qquad n_{p,j} \in Z \qquad (6)$$

where Φ is the dimensionless electric potential calculated as

$$\Phi = -\frac{FE}{RT}, \qquad (7)$$

$F$ is Faraday's constant (C/mol) and $E$ is the reduction potential of the electron (or the generic unspecified equivalent carrier) in volts.

This formulation differs from the one presented in the previous subsection in that the dimensionless potentials are now treated as additional (continuous) decision variables. The multi-objective optimization is otherwise conducted in a similar manner as described in the previous subsection, with these added variables taken into account.

Since multiple solutions exist that may return the same optimum energy yield and MDF, it is considered desirable to quantify the set of possible values of each $\Phi_j$ that is optimal. As these are continuous decision variables of a linear problem, any value of $\Phi_j$ that is between two known optimal solutions is also optimal (58). Therefore, the optimal set is a range that can be fully described by just the maximum and minimum value of $\Phi_j$. The following algorithm was devised for finding these values for each $\Phi_j$:

i.  Using Gurobi's solution pools feature, all optimal solutions that contain distinct combinations of the integer decision variables ($n_{p,j}$) are found and listed. Steps ii–v are then repeated for each of the solutions
ii. The values of $n_{p,j}$ are set at that optimal value, and the value of $F_j$ is set at the optimum MDF found. This leaves only continuous decision variables, converting the problem to a much less expensive linear program.
iii. Keeping the same energy constraint and boundaries for decision variable, the objective function is now to maximize $\Phi_j$. This returns the maximum value of $\Phi_j$ that is still optimal.

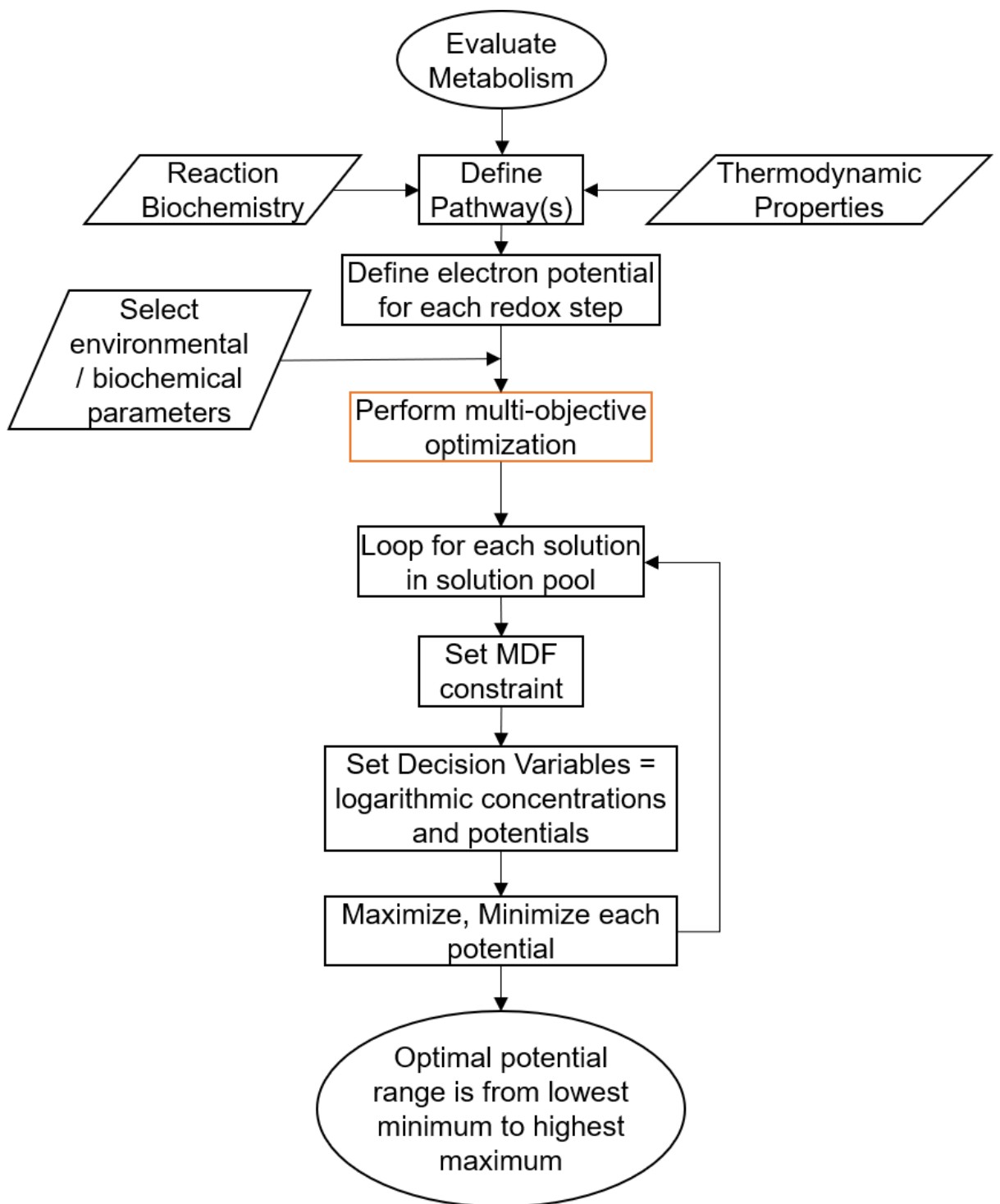

**FIG 10** Flowchart describing the steps involved in the electron carrier analysis of a given metabolic pathway. The orange block corresponds to the multi-objective optimization run in the orange box in Fig. 9.

 iv. Repeat step iii but with the objective function to minimize $\Phi_j$. This returns the minimum value of $\Phi_j$ that is still optimal.

 v. Repeat steps iii and iv for each desired $\Phi_j$

vi.   The full optimal range for any $\Phi_j$ is from the lowest minimum found in step v across the whole solution pool to the highest maximum found in step iii

A visual summary of the modified model solution strategy is presented in Fig. 10. All codes used are available online at https://doi.org/10.5281/zenodo.7635404.

## ACKNOWLEDGMENTS

This work was supported by the Sustainable Bioenergy Research Consortium under the Award No. 8434000305/EX2019-003 and by resources of the Research and Innovation Centre on $CO_2$ and $H_2$ (RICH).

A.T.: Methodology, Software, Formal analysis, Writing – original draft, Visualization. M.P.: Visualization, Writing – review & editing. J.R.: Conceptualization, Methodology, Resources, Writing – review & editing, Supervision, Project administration, Funding acquisition.

The authors declare no conflict of interest. The founding sponsors had no role in the design of the study; in the collection, analyses, or interpretation of data; in the writing of the manuscript; and in the decision to publish the results.

## AUTHOR AFFILIATION

[1]Department of Chemical and P. Engineering, Research and Innovation Centre on $CO_2$ and $H_2$ (RICH), Khalifa University, Abu Dhabi, United Arab Emirates

## AUTHOR ORCIDs

Ahmed Taha  http://orcid.org/0000-0002-9677-0733
Mauricio Patón  http://orcid.org/0000-0002-1869-4448
Jorge Rodríguez  http://orcid.org/0000-0002-5936-9676

## AUTHOR CONTRIBUTIONS

Ahmed Taha, Formal analysis, Methodology, Software, Visualization, Writing – original draft | Mauricio Patón, Visualization, Writing – review and editing | Jorge Rodríguez, Conceptualization, Funding acquisition, Methodology, Project administration, Resources, Supervision, Writing – review and editing

## DATA AVAILABILITY

The data sets generated during and/or analyzed during the current study are generated by the code used and available online at https://doi.org/10.5281/zenodo.7635404.

## ADDITIONAL FILES

The following material is available online.

### Supplemental Material

**Supplemental Material (mSystems01274-24-s0001.pdf).** Pathway details, Tables S1 to S3, and Fig. S1 to S3.

### Open Peer Review

**PEER REVIEW HISTORY (review-history.pdf).** An accounting of the reviewer comments and feedback.

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
