## [Reviewer comments · mSystems]

Bioenergetic trade-offs can reveal the path to superior microbial CO₂ fixation pathways

Ahmed Taha, Mauricio Patón, and Jorge Rodriguez

Corresponding Author(s): Jorge Rodriguez, Khalifa University

Review Timeline:

Submission Date:	September 24, 2024
Editorial Decision:	November 6, 2024
Revision Received:	November 18, 2024
Accepted:	December 17, 2024

Editor: Christopher Marshall

Reviewer(s): Disclosure of reviewer identity is with reference to reviewer comments included in decision letter(s). The following individuals involved in review of your submission have agreed to reveal their identity: Christopher E Lawson (Reviewer #1)

Transaction Report:

DOI: <https://doi.org/10.1128/msystems.01274-24>

Re: mSystems01274-24 (Bioenergetic trade-offs can reveal the path to superior microbial CO₂ fixation pathways)

Dear Dr. Jorge Rodriguez:

Overall, the reviewers were enthusiastic about the work presented and suggested mostly minor modifications. I agree with the sentiment of the reviewers that the results and analyses could be presented clearer so that a general scientific audience can grasp the methodology and importance of the work.

Revision Guidelines

Sincerely,
Christopher Marshall
Editor
mSystems

Reviewer #1 (Comments for the Author):

This study is a comprehensive and compelling investigation of efficiency and rate of existent and hypothetical biological carbon

fixation pathways. The results that aerobic carbon fixation is less energetically efficient than anaerobic ones, and that engineered pathways have small or negative MDFs despite large overall free energy changes, are especially interesting. The manuscript's clarity should be improved. Particularly: 1) the definition of ATP cost and how it relates to proton translocation, 2) the mechanism by which protons are traded for driving force in the model and its biological relevance, 3) the way in which ATP yield and MDF are simultaneously optimized (presumably they are optimized sequentially). While the work is very compelling, it should be communicated more clearly to readers.

Specific Comments:

- Line 68: Why is the trade-off between rate and yield especially important in anaerobic systems? Is it because both are intrinsically lower in the absence of oxygen? This should be articulated more clearly.
- Line 87: Electrical potential should be changed to reduction potential for clarity. Electrical potential is amount of work needed to move charge and can't be assigned to a particular electron carrier in isolation. A reduction potential versus a standard hydrogen electrode can.
- Line 110: Flux would be a better terminology than rate, since J_+ and J_- are in moles per volume per time. Perhaps this should be rectified throughout the paper.
- Line 113: It should be made clear that G_{diss} is the inverse of the reaction gibbs free energy change (i.e., positive when the forward reaction is spontaneous).
- Line 117: Typo ("the is" should be "the reaction is").
- Line 151: The definition of net ATP cost and how it was obtained is unclear. Was an initial ATP cost assigned, then the maximal number of proton translocations per mole CO_2 found and subtracted from that initial ATP cost? These details should be addressed briefly before the results, otherwise interpretation is difficult.
- Line 159: "more efficient energy-wise" is a bit vague. Could have many interpretations, but I assume that what is meant is that less ATP is required per mole of fixed carbon? This should be rephrased.
- Line 171 to 172: The statement that aerobic pathways will not be running under conditions of no imposed MDF does not make sense. The imposed MDF is a simulation parameter, the pathway's activity or inactivity is not an output of the simulation but an empirical fact. This sentence should be reworked.
- Line 187: The connection between the tightness of contour plot lines and energy recovery efficiency should be elaborated upon briefly.
- Line 200: What does trading protons in exchange for driving force mean? Is it a parameter of proton translocation reactions in the model? It can be imagined that reducing the required proton translocations would make these reactions more exergonic at the cost of ATP generation, which could have knock-on effects throughout the metabolic network and increase MDF. Whether or not this is the correct reasoning, the manuscript does not make this clear. Moreover, what is the biological relevance of allowing fewer protons to be translocated in exchange for more energy dissipation. Is there empirical evidence for this phenomenon?
- Figure 3: Overlap between MDF lines (dashed) hinders legibility. Maybe different line styles could be used, or different opacities?
- Line 235: Is this because a different reaction apart from the proton translocation is the bottleneck? This should be clarified.
- Line 244: Addressing the comments above should also improve the clarity of the results in subsequent sections.

Reviewer #2 (Comments for the Author):

The work by Taha et al. presents a computational study of the bioenergetic trade-offs between growth rate and energy efficiency in microbial CO_2 fixation pathways. The study finds that anaerobic pathways, particularly the reverse tricarboxylic acid cycle (rTCA) and Wood-Ljungdahl pathways, demonstrate superior energy efficiency across diverse environmental conditions. This could inform metabolic engineering strategies for CO_2 fixation in industrial microbial applications. Additionally, the authors investigate optimal electron carrier potentials, offering insights into the selection of electron carriers in metabolic reactions based on thermodynamic efficiency.

Metabolic engineering for improved CO_2 fixation efficiencies in microbial systems is a field of growing interest. The authors apply here a solid and rigorous computational methodology that they previously developed to this very relevant topic, which is the main strength of the paper. The work seeking to find the optimal electron carrier potentials is very innovative and interesting, and suggests first-principles criteria for the selection of electron carriers in metabolic engineering. The methodology is technically sound and seems correct to me. Overall, I think the paper is thus a very valuable contribution, and I do recommend its publication after some minor revisions detailed below.

My main suggestion for improvement is regarding the clarity of the presentation and the explanations in the results section. The computational methodology is quite complex, and the figures as presented right now are very information dense. Making an effort to walk the reader by the hand through both the method and the results might benefit the work substantially by expanding the potential readership, and increasing the impact. Some specific suggestions are:

- Figures 1, 2, and 4, compare different CO_2 fixation pathways in their ATP yield, for a range of concentrations of CO_2 and H_2 . However, the comparison of the color gradients between the panels is visually very hard. For example, the authors could add

barplots comparing the ATP yield of the different pathways at a couple of relevant CO₂ and H₂ concentration, helping the reader to quickly grasp the message that authors are trying to get through.

- Figure 3 was challenging at first, and I could grasp it only after going to the Methods section and seeing how it is done (the epsilon method). I believe the reader should be able to understand precisely what is being done even without understanding exactly the technical details of how it is done. So a clearer explanation would be appreciated in the results section.

- The analysis in Fig. 3 (and the analogous analysis of postulated pathways) would gain from a clearer explanation why in some pathways (e.g. rTCA or WL), the driving force remains high when more protons are traded for MDF (e.g. in rTCA, for 3 CPTs, driving force is always between 4 and 5 kJ/mol approx, while for 0 CPTs is between 0 and 1 approx), while in some other pathways MDF always goes to zero when the energy is recovered as ATP. I suspect this has to do with pathway stoichiometry and at which steps the proton translocation occurs, but I do not fully understand it from the text as is now. Illustrating somehow the solutions that the optimizer finds, akin to Fig. S3, would be great.

- The same request for more clarity and transparency in the explanation applies to the analysis in Fig 6, which I think is the most innovative part of this paper. I would make an effort to explain better what type of optimization has been done. My advice is that if the explanation and clarity requires, do not shy away from showing some equations also in the results section (even if then the Methods section repeats them and fleshes out how the analysis has been done). Also, I do not fully understand how the permitted carriers shown in the panel subtitles have been determined, since I do not see overlap with the colour-shaded regions in all cases.

- Overall, my personal belief is that the paper would gain by just showing the results for the natural pathways (but walking the reader by the hand through all the analyses), and perhaps leaving the postulated pathways for the supplement. But this is of course completely up to the authors to decide.

- L117 - word missing "the ... is away from eq"

- The wording in the discussion could be chosen better at points, e.g. L369-370, what does it exactly mean that "the trade-off is not constant and nonlinear"? The sentence in L384-386 is quite wordy as well.

Reviewer #3 (Comments for the Author):

I was delighted to have the opportunity to review this very interesting manuscript. It is a compelling and informative theoretical analysis of different carbon fixation strategies. More broadly it is an example of how we can do excellent microbiology with a pen (or a pen and a computer) rather than a pipette.

Thus although the body of the paper considers only the question ostensibly in hand, an analysis of the the energetics of carbon fixation, the methods used to make this analysis are just as interesting (to me at least). Which is not to deny the strategic importance of a detailed understanding of carbon fixation by both known and hypothetical pathways.

The manuscript is well written and (as far as I can see) has almost no errors and the software has been made available.

I have the following minor comments. These are mostly matters of judgement and the authors may disagree.

Line 101 "At least six main pathways" It might be helpful to bring figures 7 and 8 forward to this point. I was unfamiliar with some of the pathways and with the details of all of them. This hampered my understanding of the paper.

Line 117 "FFE" This acronym is not defined at this point in the manuscript.

Line 124 "highly efficient computational methodology by [44]." This innovation lies at the heart of the paper. It might be worth adding a sentence on the mathematical basis of these advances.

Figure 3. This figure is too small. The very fine lines are hard to follow.

Line 231 "The figure" Which figure is this?

Line 289 " This severely restricts" An excellent point.

Figure 5. This figure is too small. The very fine lines are hard to follow.

Line 376 "irreducible inconsistent subsets" This is not a text book mathematical concept (well not in my math textbooks anyway) and a reference and one line explanation would be helpful.

Table 1. Please define SLP in the table legend.

The authors would like to thank the reviewers and editor for the time and effort in reviewing the manuscript. We have found all the comments useful and valuable and we believe they have contributed to increasing the quality of the revised manuscript. Below we provide detailed responses to the changes suggested.

Editor:

Overall, the reviewers were enthusiastic about the work presented and suggested mostly minor modifications. I agree with the sentiment of the reviewers that the results and analyses could be presented clearer so that a general scientific audience can grasp the methodology and importance of the work.

We thank the editor and reviewers for the valuable feedback on the manuscript. We have incorporated relevant changes towards addressing the clarity concerns and as well as all other comments.

Reviewer 1:

This study is a comprehensive and compelling investigation of efficiency and rate of existent and hypothetical biological carbon fixation pathways. The results that aerobic carbon fixation is less energetically efficient than anaerobic ones, and that engineered pathways have small or negative MDFs despite large overall free energy changes, are especially interesting. The manuscript's clarity should be improved. Particularly: 1) the definition of ATP cost and how it relates to proton translocation, 2) the mechanism by which protons are traded for driving force in the model and its biological relevance, 3) the way in which ATP yield and MDF are simultaneously optimized (presumably they are optimized sequentially). While the work is very compelling, it should be communicated more clearly to readers.

Specific Comments:

- Line 68: Why is the trade-off between rate and yield especially important in anaerobic systems? Is it because both are intrinsically lower in the absence of oxygen? This should be articulated more clearly.

The trade-off is especially important for microbial processes with low Gibbs energy yield as stated in line 68. Anaerobic digestion is simply one example of such a system where the absence of a strong electron acceptor (oxygen) leads to low Gibbs metabolisms.

- Line 87: Electrical potential should be changed to reduction potential for clarity. Electrical potential is amount of work needed to move charge and can't be assigned to a particular electron carrier in isolation. A reduction potential versus a standard hydrogen electrode can.

This term was indeed misleading and has been corrected to 'reduction potential'.

- Line 110: Flux would be a better terminology than rate, since $J+$ and $J-$ are in moles per volume per time. Perhaps this should be rectified throughout the paper.

The definition of rate in most biochemical contexts is in volumetric units (i.e. moles reacted per time per volume) and we opted for the rate term as it is in this case synonymous.

- Line 113: It should be made clear that G_{diss} is the inverse of the reaction gibbs free energy change (i.e., positive when the forward reaction is spontaneous).

An additional statement was added in line 113 to clarify this and remove any ambiguity.

- Line 117: Typo ("the is" should be "the reaction is").

We thank the reviewer for pointing this error. The missing word (reaction) has been added.

- Line 151: The definition of net ATP cost and how it was obtained is unclear. Was an initial ATP cost assigned, then the maximal number of proton translocations per mole CO_2 found and subtracted from that initial ATP cost? These details should be addressed briefly before the results, otherwise interpretation is difficult.

The net ATP cost is indeed the sum of the ATP from SLP plus that from proton translocations calculated via the optimisation solver. These details are clarified in the methods section lines 479-506. We agree that ideally the methods section could precede the results and discussion section for clarity before. This order was however imposed by the journal format rules.

- Line 159: "more efficient energy-wise" is a bit vague. Could have many interpretations, but I assume that what is meant is that less ATP is required per mole of fixed carbon? This should be rephrased.

This is indeed the case, and the sentence has been rephrased to clarify this message and remove any ambiguity.

- Line 171 to 172: The statement that aerobic pathways will not be running under conditions of no imposed MDF does not make sense. The imposed MDF is a simulation parameter, the pathway's activity or inactivity is not an output of the simulation but an empirical fact. This sentence should be reworked.

We acknowledge the difficulty in correctly interpreting this sentence; it has been rephrased to clarify the correct message (an aerobic C-fixing microbe is unlikely to prioritise yield over rate).

- Line 187: *The connection between the tightness of contour plot lines and energy recovery efficiency should be elaborated upon briefly.*

This connection has been explored in lines 258 and 354. The higher the density of proton-translocating reactions in the pathway, the more likely it will be able to recover any additional Gibbs yield (from improving environmental concentrations or CPTs traded) in the form of driving forces all across the metabolic network.

- Line 200: *What does trading protons in exchange for driving force mean? Is it a parameter of proton translocation reactions in the model? It can be imagined that reducing the required proton translocations would make these reactions more exergonic at the cost of ATP generation, which could have knock-on effects throughout the metabolic network and increase MDF. Whether or not this is the correct reasoning, the manuscript does not make this clear. Moreover, what is the biological relevance of allowing fewer protons to be translocated in exchange for more energy dissipation. Is there empirical evidence for this phenomenon?*

The details of the approach are explained in the methods section. The number of protons translocated by each membrane-bound enzyme bound reaction in the pathway is a decision variable in the optimisation problem. It indeed has a strong effect on the network, as an investment of CPTs in an earlier reaction step could push some subsequent reactions in the realm of feasibility Gibbs-wise, allowing for a downstream recovery of a greater amount of CPTs than the number invested. The optimisation is multi-objective with two competing objectives (maximizing the minimum driving force and the yield of ATP), with the tradeoff between the two objectives forming a pseudo-Pareto front as detailed in [1]. The biological relevance of having more energy dissipation is an increase in effective enzyme utilisation, thus increasing the kinetic rate.

- Figure 3: *Overlap between MDF lines (dashed) hinders legibility. Maybe different line styles could be used, or different opacities?*

In response to this and the other reviewers, we have remade figures 3 and 5 into a stacked plot for clarity, thus completely separating the MDF and the ATP cost lines. We agree that this was probably needed to improve the graph's readability.

- Line 235: *Is this because a different reaction apart from the proton translocation is the bottleneck? This should be clarified.*

An additional statement has been added to clarify this observation further L243-245.

- Line 244: *Addressing the comments above should also improve the clarity of the results in subsequent sections.*

Indeed, the above comments and those from the other reviewers have led to several improvements in the results and discussion aiming at a more clear message and readability.

Reviewer 2:

The work by Taha et al. presents a computational study of the bioenergetic trade-offs between growth rate and energy efficiency in microbial CO₂ fixation pathways. The study finds that anaerobic pathways, particularly the reverse tricarboxylic acid cycle (rTCA) and Wood-Ljungdahl pathways, demonstrate superior energy efficiency across diverse environmental conditions. This could inform metabolic engineering strategies for CO₂ fixation in industrial microbial applications. Additionally, the authors investigate optimal electron carrier potentials, offering insights into the selection of electron carriers in metabolic reactions based on thermodynamic efficiency.

Metabolic engineering for improved CO₂ fixation efficiencies in microbial systems is a field of growing interest. The authors apply here a solid and rigorous computational methodology that they previously developed to this very relevant topic, which is the main strength of the paper. The work seeking to find the optimal electron carrier potentials is very innovative and interesting, and suggests first-principles criteria for the selection of electron carriers in metabolic engineering. The methodology is technically sound and seems correct to me. Overall, I think the paper is thus a very valuable contribution, and I do recommend its publication after some minor revisions detailed below.

My main suggestion for improvement is regarding the clarity of the presentation and the explanations in the results section. The computational methodology is quite complex, and the figures as presented right now are very information dense. Making an effort to walk the reader by the hand through both the method and the results might benefit the work substantially by expanding the potential readership, and increasing the impact. Some specific suggestions are:

- Figures 1, 2, and 4, compare different CO₂ fixation pathways in their ATP yield, for a range of concentrations of CO₂ and H₂. However, the comparison of the color gradients between the panels is visually very hard. For example, the authors could add barplots comparing the ATP yield of the different pathways at a couple of relevant CO₂ and H₂ concentration, helping the reader to quickly grasp the message that authors are trying to get through.

We agree that the ATP cost plots are dense in information and require effort to be understood. We have attempted several different plot types to visualise this information, including 3D surfaces and elevation plots. After evaluating those options we opted for the current format (2D filled contour plot) that, though not perfect, we found it, arguably, the most clear to present this information. In addition, we want to emphasize that the purpose of these figures is not an accurate reading of exact ATP cost values of C fixation for some [CO₂] and [H₂], but on the representation of trends and sensitivity to concentration changes and how this differs for different pathways.

- Figure 3 was challenging at first, and I could grasp it only after going to the Methods section and seeing how it is done (the epsilon method). I believe the reader should be able to

understand precisely what is being done even without understanding exactly the technical details of how it is done. So a clearer explanation would be appreciated in the results section.

We have remade figure 3 using a stacked plot approach, thus completely separating the MDF and the ATP cost lines. We believe this significantly improves the graph's readability. We agree that ideally the methods section could precede the results and discussion section for clarity before. We understood that this order was however imposed by the journal format rules.

- The analysis in Fig. 3 (and the analogous analysis of postulated pathways) would gain from a clearer explanation why in some pathways (e.g. rTCA or WL), the driving force remains high when more protons are traded for MDF (e.g. in rTCA, for 3 CPTs, driving force is always between 4 and 5 kJ/mol approx, while for 0 CPTs is between 0 and 1 approx), while in some other pathways MDF always goes to zero when the energy is recovered as ATP. I suspect this has to do with pathway stoichiometry and at which steps the proton translocation occurs, but I do not fully understand it from the text as is now. Illustrating somehow the solutions that the optimizer finds, akin to Fig. S3, would be great.

We did indeed notice this phenomenon and originally wanted to include it in the discussion. To investigate it we did dig within the pathway intermediate concentrations and individual reaction CPTs, like in Fig S3. to search for a clear cause. However, the cause of the phenomenon remains non obvious to us and it is likely to be due to a non-trivial and composite effect of several factors, perhaps relating to the molarity of the electron carrier regeneration reaction as well as the occurrence of some intermediate oxidation steps in these two pathways.

- The same request for more clarity and transparency in the explanation applies to the analysis in Fig 6, which I think is the most innovative part of this paper. I would make an effort to explain better what type of optimization has been done. My advice is that if the explanation and clarity requires, do not shy away from showing some equations also in the results section (even if then the Methods section repeats them and fleshes out how the analysis has been done). Also, I do not fully understand how the permitted carriers shown in the panel subtitles have been determined, since I do not see overlap with the colour-shaded regions in all cases.

The word 'permitted' in this context means electron carriers that have been reported in biochemical literature (KEGG, biocyc etc...) to be used in a particular reaction step i.e. having an enzyme whose active site fits that combination of substrate and electron carrier. On the other hand, the figure should be used to read which electron carrier would be optimal, from a purely bioenergetic (both ATP cost and driving force) perspective. We have significantly expanded Figure 6's caption to remind the reader of the optimisation steps performed to generate the figure.

- Overall, my personal belief is that the paper would gain by just showing the results for the natural pathways (but walking the reader by the hand through all the analyses), and perhaps

leaving the postulated pathways for the supplement. But this is of course completely up to the authors to decide.

We indeed thought about organising the paper as suggested, however we wanted to emphasize the value of the work in its application to bottom-up approaches in microbial metabolic engineering which could be based on postulated pathways, that is why eventually we kept this section in the main text.

- L117 - word missing "the ... is away from eq"

We thank the reviewer for spotting this error. The missing word (reaction) has been added.

- The wording in the discussion could be chosen better at points, e.g. L369-370, what does it exactly mean that "the trade-off is not constant and nonlinear"? The sentence in L384-386 is quite wordy as well.

A rewrite of several sentences in the discussion (including the phrase in line 369-370) was conducted in order to improve clarity acknowledging that text is at times on the lengthy side due to the complexity of the discussion.

Reviewer 3:

I was delighted to have the opportunity to review this very interesting manuscript. It is a compelling and informative theoretical analysis of different carbon fixation strategies. More broadly it is an example of how we can do excellent microbiology with a pen (or a pen and a computer) rather than a pipette.

Thus although the body of the paper considers only the question ostensibly in hand, an analysis of the the energetics of carbon fixation, the methods used to make this analysis are just as interesting (to me at least). Which is not to deny the strategic importance of a detailed understanding of carbon fixation by both known and hypothetical pathways.

The manuscript is well written and (as far as I can see) has almost no errors and the software has been made available.

The authors would like to thank the reviewer for the highly positive review. We believe that such computational analysis based on fundamental principles (thermodynamics and kinetics) can offer large insight into the development of hypothetical and industrially efficient carbon fixing metabolic pathways.

I have the following minor comments. These are mostly matters of judgement and the authors may disagree.

Line 101 "At least six main pathways" It might be helpful to bring figures 7 and 8 forward to this point. I was unfamiliar with some of the pathways and with the details of all of them. This hampered my understanding of the paper.

We have made more explicit in lines 102-104 where the full details of the pathways can be found.

Line 117 "FFE" This acronym is not defined at this point in the manuscript.

The acronym 'FFE' was already defined in line 111.

Line 124 "highly efficient computational methodology by [44]." This innovation lies at the heart of the paper. It might be worth adding a sentence on the mathematical basis of these advances.

We have added a sentence in line 129-131 in line with that.

Figure 3. This figure is too small. The very fine lines are hard to follow.

We have remade figure 3 and 5 using a stacked plot approach, thus completely separating the MDF and the ATP cost lines. We believe this significantly improves the graph's readability as suggested by the reviewer.

Line 231 "The figure" Which figure is this?

A reference to the figure number (3) was added to the sentence.

Line 289 " This severely restricts" An excellent point.

The authors thank the reviewer for their positive response.

Figure 5. This figure is too small. The very fine lines are hard to follow.

We have remade figure 3 and 5 using a stacked plot approach, thus completely separating the MDF and the ATP cost lines. We agree this improves the graph's readability.

Line 376 "irreducible inconsistent subsets" This is not a text book mathematical concept (well not in my math textbooks anyway) and a reference and one line explanation would be helpful.

A line was added that defines what an irreducible inconsistent subset is, along with a reference to a paper detailing how it can be obtained using a Simplex-type optimisation algorithm.

Table 1. Please define SLP in the table legend.

This was indeed an overlook. The abbreviation definition has been added to the table legend and in the first time it is used in the paper.

References

1. Taha A, Patón M, Penas DR, Banga JR, Rodríguez J. Optimal evaluation of energy yield and driving force in microbial metabolic pathway variants. *PLOS Computational Biology*. 2023;19: e1011264. doi:10.1371/journal.pcbi.1011264

Re: mSystems01274-24R1 (Bioenergetic trade-offs can reveal the path to superior microbial CO₂ fixation pathways)

Dear Dr. Jorge Rodriguez:

Congratulations! Your manuscript has been accepted, and I am forwarding it to the ASM production staff for publication. Your paper will first be checked to make sure all elements meet the technical requirements. ASM staff will contact you if anything needs to be revised before copyediting and production can begin. Otherwise, you will be notified when your proofs are ready to be viewed.

Sincerely,
Christopher Marshall
Editor
mSystems

Reviewer #1 (Comments for the Author):

This work by Taha et al. is an innovative and exciting effort towards understanding optimality in natural and engineered metabolism. The fundamental trade-off between yield and rate in metabolism has been the subject of much discussion in recent years. Understanding how catabolism can favour one over the other, and understanding which conditions are selective for rate over yield and vice-versa, may help address many open questions in biology (from overflow metabolism to cancer progression), and may be informative to the design of engineered organisms for diverse technological applications.

This manuscript addresses the trade-off between yield and rate in a technologically important metabolism, carbon-fixation, with a novel methodology which evaluates a multitude of pathway variants. Their results are novel and potentially impactful, particularly that aerobic pathways may favour rate while anaerobic ones seem to favour yield, and that engineered pathways would likely have low rates despite large overall free energy changes. The authors gracefully addressed previous comments and have improved the communication of their results. I have no further comments/suggestions

Reviewer #2 (Comments for the Author):

While I might have approached certain aspects of the presentation differently if I were the author, I am not, and these choices are ultimately up to the authors. That said, I find this to be a rigorously conducted and significant piece of research. Therefore I recommend its publication.

Reviewer #3 (Comments for the Author):

I have read the revised manuscript with considerable satisfaction, and I have no further comments to make, except to apologise for the lateness of my review.